# Career Path Modeling and Recommendations with Linkedin Career Data and Predicted Salary Estimations

**Micaela T. Cerilla**[*]**, Aaron A. Santillan**[*]**, & Carl John R. Viñas**[*]
Department of Computer Science
Polytechnic University of The Philippines
`{mtcerilla,aasantillan,cjrvinas}@iskolarngbayan.pup.edu.ph`

**Michael B. Dela Fuente**
Department of Computer Science
Polytechnic University of The Philippines
`mbdelafuente@pup.edu.ph`

## Abstract

Career planning involves devising a sequence of steps that build up an ideal career path for a person. However, career planning has become more complex in recent years, demanding the need for better models and systems for recommending Career Paths. With that in mind, we explored new variables and techniques that could help in predicting better career paths. We built a Long-Short Term Memory Network with Self Attention Layers called LSTM-ATT, that predicts a person's possible career path using Linkedin Career History and new variables such as salary estimations and social networks. We measured the model's performance in terms of Mean Percentile Rank and Precision at 50 and 100. We found that LSTM and self-attention layers were able to show good predictive performance for multi-class classification even with over 6000 classes for companies and skills, effectively beating a multi-channel CNN for all metrics. However, by checking the versions of either model with added features, they did not yield any major increase in predictive accuracy against the models without it. This leads us to conclude that the added variables did not help in predicting better career paths.

## 1 Introduction

We define a career path as a series of steps that allow individuals to achieve a specific goal or objective. There have been a few studies exploring predicting and recommending career paths (Wan & Ye, 2022; Natividad et al., 2019; Li et al., 2017b;a; Ghosh et al., 2020; Zhang et al., 2020).

However, choosing a career path has become more complex in recent years because of the following causes: (1) inaccessibility to career counseling, (2) the diversity of roles that an individual may partake in, and (3) career planning can lead to information overload (Wan & Ye, 2022). Given the limitations of existing models, we explored new variables and techniques that could help in predicting better career paths. We look into unexplored variables from previous studies (Ghosh et al., 2020), namely social networks (better known as LinkedIn recommendations) and predicted salary estimations while utilizing Long-Short Term Memory Networks and Self-Attention Layers for improved accuracy.

## 2 Problem Context and Experimental Setup

Predicting a career path through machine learning can be defined as two recommendation problems:

---

[*]All three authors listed here provided equal contributions to the creation of this research and paper. The ordering was decided through the probability of a D20 dice roll.

1. The first problem is defined as recommending the user's **latest** job and company as a relevant item.

2. The second problem is defined as recommending all the skills that an individual possesses in the given user or Linkedin Profile as all relevant items.

To know if LSTM-ATT and the added features can improve career path recommendations, we answered both recommendation problems above through the following experimental process. We created different versions of LSTM-ATT models and M-CNN (another algorithm used for career path recommendations) with or without the added variables/features, trained all versions of both models, optimized LSTM-ATT through hyperparameter tuning, and evaluated all models' accuracy in recommendations through the use of the usual metrics in the domain of recommendation systems, namely the Mean Percentile Rank (MPR)and Precision @ K (at 10 and 100). We then compared both models without the features if there are any major changes in accuracy and evaluated if adding the new features/variables resulted in a major change in recommendation accuracy. Through this process, we can understand if both solutions helped in predicting better career paths.

## 3 MODEL ARCHITECTURE

We developed a model called LSTM-ATT and used scraped data obtained from LinkedIn and Payscale to predict career paths (See Section 5.1). For Linkedin, we extracted users' career histories (which consist of all the jobs and companies they transitioned in their professional journey), set of skills, education history, and Linkedin recommendations posted by the user. On the other hand, predicted salaries per job are derived by matching jobs from each job history of all users to the predicted salaries of each job taken from Payscale. We then converted all features into embeddings, concatenated them, and fed them to the LSTM-ATT model to predict the latest career path.

## 4 RESULTS, DISCUSSIONS, AND RECOMMENDATIONS

Based on the predictive results of the LSTM-ATT models below (See Table 1), LSTM and self-attention layers were able to show good predictive performance for multi-class classification even with over 6000 classes for companies and skills, effectively beating a multi-channel CNN for all metrics. This leads us to conclude that LSTM and Attention Layers significantly helped in performance. Using the same table, by checking the versions of either model with added features, it can be seen that they did not yield in any major increase in predictive accuracy against the models without it. This leads us to conclude that the added variables did not help in predicting better career paths.

For future work, further investigation is needed in checking if there is a change in results with bigger datasets since the size of our dataset used in our experiment pales in comparison to previous works(Ghosh et al., 2020).

Table 1: Results on Model Predictions for Jobs, Companies, and Skills for LSTM-ATT and M-CNN

| Methods | Jobs | | | Companies | | | Skills | | |
|---|---|---|---|---|---|---|---|---|---|
| | MPR | P@10 | P@100 | MPR | P@10 | P@100 | MPR | P@10 | P@100 |
| LSTM-ATT | 0.011 | 0.107 | 0.134 | 0.118 | 0.747 | 0.952 | 0.059 | 0.293 | 0.402 |
| LSTM-ATT + SAL | 0.017 | 0.088 | 0.113 | 0.111 | 0.760 | 0.940 | 0.062 | 0.299 | 0.407 |
| LSTM-ATT + SN | 0.017 | 0.128 | 0.173 | 0.063 | 0.854 | 0.913 | 0.061 | 0.286 | 0.395 |
| LSTM-ATT + SAL + SN | 0.021 | 0.105 | 0.163 | 0.063 | 0.086 | 0.921 | 0.060 | 0.285 | 0.379 |
| M-CNN | 0.012 | 0.074 | 0.090 | 0.339 | 0.446 | 0.592 | 0.059 | 0.212 | 0.274 |
| M-CNN + SAL | 0.011 | 0.049 | 0.069 | 0.030 | 0.371 | 0.542 | 0.062 | 0.229 | 0.293 |
| M-CNN + SN | 0.004 | 0.037 | 0.038 | 0.048 | 0.120 | 0.197 | 0.056 | 0.198 | 0.251 |
| M-CNN + SAL + SN | 0.047 | 0.047 | 0.060 | 0.014 | 0.165 | 0.498 | 0.058 | 0.221 | 0.267 |

Definitions: *LSTM-ATT = LSTM with Self Attention Layers, M-CNN = Multi-Channel CNN, SAL = Salary Feature, SN = Social Networks*

## 5  APPENDICES

### 5.1  VISUALIZATION OF THE MODEL ARCHITECTURE AND OVERALL PROCESS

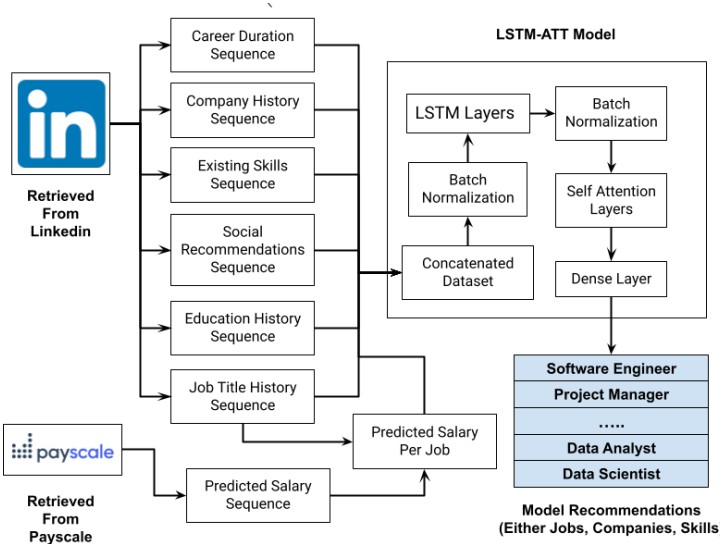

### 5.2  EXPERIMENT PARAMETERS

For generating the experimental results for both models (Table 1), Here are the defined parameters for the entire experiment. All the embedding dimensions and epoch parameters for LSTM-ATT were selected through hyperparameter tuning.

1. **For LSTM-ATT**
   (a) Skills Embedding Output Dim - 480
   (b) Jobs Embedding Output Dim - 128
   (c) Establishments Embedding Output Dim - 384
   (d) Education Embedding Output Dim - 128
   (e) Salaries Embedding Output Dim - 128
   (f) Duration Embedding Output Dim - 128
   (g) Given Recommendations Embedding Output Dim - 128
   (h) Received Recommendations Embedding Output Dim - 128
   (i) Epochs - 30
   (j) Learning Rate - 0.001
   (k) Optimizer - Adam
   (l) Verbose - 2
   (m) Loss Function - modified Sparse Categorical CrossEntropy

2. **For M-CNN**
   (a) Skills Embedding Output Dim - 480
   (b) Jobs Embedding Output Dim - 128
   (c) Establishments Embedding Output Dim - 384
   (d) Education Embedding Output Dim - 128
   (e) Salaries Embedding Output Dim - 128
   (f) Duration Embedding Output Dim - 128
   (g) Given Recommendations Embedding Output Dim - 128
   (h) Received Recommendations Embedding Output Dim - 128
   (i) Epochs - 30

    (j) Learning Rate - 0.001

    (k) Optimizer - Adam

    (l) Verbose - 2

    (m) Loss Function - modified Sparse Categorical CrossEntropy

## 5.3 Used Materials

Listed here are all resources we used for this research.

1. Code Repository - contains both the actual code and datasets.

2. Notebooks - the actual code to run the model and generate the experiments.

3. Datasets - contains the pre-processed dataset for training, evaluating, and hyper-parameter tuning the model. Also includes the required item dictionaries for forming the tokenizers and embeddings needed for the model.

# 6 URM Statement

The authors acknowledge that at least one key author of this work meets the URM criteria of ICLR 2023 Tiny Papers Track.

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
