# OpenReview forum: "Career Path Modeling and Recommendations with Linkedin Career Data and Predicted Salary Estimations"
_ICLR.cc/2023/TinyPapers — Submitted to Tiny Papers @ ICLR 2023_

### Official Review · Reviewer_7qXV · 2023-03-18

**Confidence:** 5

**Summary Of Contributions:**

LSTM-ATT performance with different feature combinations

**Rating:**

Great Start (GS): a submission which meets some of the reviewing criteria but has room for improvement

**Strengths And Weaknesses:**

The paper focuses on performance of only LSTM-ATT with additional features(latest job & skills)
Considering one of the authors meets URM criteria, this is a good start.
Scope for improvement would be focus on optimization of LSTM-ATT and for future compare the performance of LSTM-ATT & Hybrid CNN used in mentioned reference paper Wan& Ye,2022


**Suggested Changes:**

Request both authors to discuss and improve their paper with more research
1. Authors can also focus on optimization of the LSTM-ATT with hyperparameter tuning and compare performance.
2. For future papers, Ideal comparison would be performance two or more model techniques rather than just features selected.

---

> ### Author Response · Authors · 2023-06-01
> **Added requested changes by reviewer 7qXV**
>
> Thank you all reviewers for your inciteful and positive feedback and recommendations.  We addressed the mentioned issues and concerns below:
>
> 1. Concern: Authors can also focus on the optimization of the LSTM-ATT with hyperparameter tuning and compare performance.
>     - Author's Action: We failed to clarify in the paper, but the versions of LSTM-ATT used for generating the experimental results written in this paper, were already optimized through hyperparameter tuning. To resolve this confusion, we added details on the parameters used for our model and the notebooks used for creating the experimental results written on the paper in the appendix section.
>
>
> 2. Concern: For future papers, Ideal comparison would be the performance of two or more model techniques rather than just features selected.
>    - Author's Action: We failed to add it to the paper due to the page limit, but we have compared our work against other techniques upon conducting this research. To resolve this, We added the results and conclusions of our model (LSTM-ATT) being compared against M-CNN (multi-channel CNN), a model similar to the work of Wan& Ye,2022.

---

### Meta-Review · Area_Chair_2Sfn · 2023-04-07

**Recommendation:** Invite to archive
**Confidence:** 4

**Metareview:**

1. LSTM-ATT performance based on feature selection is clear and crisp
2. The paper provides architecture, and results of their research
3. The paper though well explained fails to provide code or data to make the findings reproducible
4. The submission follows basic requirements


**Summary:**

LSTM-ATT performance based on feature selection, reproducible code missing

**Comments And Feedback To The Authors:**

Considering one of the author fall under URM category this is a good start.
There are few changes that must be added for the paper to be considered as CCR.
1. *__Reproducibility__* review criteria:  Include links to code and data from your research.
2. Provide expansion of the terms in __results table__ for terms like  LSTM-ATT-SAL, LSTM-ATT-SN, LSTM-ATT-SAL-SN. Provide explanation below the results table
3. Ideal to have performance comparison with more than one techniques(LSTM-ATT used currently)

Optional change:
Time permits authors can venture into optimization with hyperparameter tuning.


**Reason For Not Giving A Higher Recommendation:**

1. Reproducible code & data missing
2. Performance compared with only one method LSTM-ATT based on different features
3. Optimization hyperparameter tuning or performance comparison with another Machine learning method apart from only LSTM-ATT would have boosted this paper for higher recommendations

**Reason For Not Giving A Lower Recommendation:**

N/A

---

> ### Author Response · Authors · 2023-06-01
> **Added Requested Revisions by Area Chair 2Sfn**
>
> Thank you all reviewers for your inciteful and positive feedback and recommendations. We addressed the mentioned issues and concerns below:
>
> 1. Concern: Reproducibility review criteria: Include links to code and data from your research.
>     - Author's Action:  To resolve this issue, we added in the appendix all the resources we used for this research (datasets and jupyter notebooks) that can be accessed via GitHub and the model parameters we used for this study.
>
> 2. Concern: Provide expansion of the terms in results table for terms like LSTM-ATT-SAL, LSTM-ATT-SN, LSTM-ATT-SAL-SN. Provide explanation below the results table
>     - Author's Action: As requested, we added the applicable terms for all parts used in the table for the results and discussion section.
>
> 3. Concern: Ideal to have performance comparison with more than one techniques(LSTM-ATT used currently)
>     - Author's Action: We failed to add it to the paper due to the page limit, but we have compared our work against other techniques upon conducting this research. To resolve this, We added the results and conclusions of our model (LSTM-ATT) being compared against M-CNN (multi-channel CNN), a model similar to the work of Wan& Ye,2022.
>
> 4. Concern: Optional change: Time permits authors can venture into optimization with hyperparameter tuning.
>     - Author's Action:  We failed to clarify in the paper, but the versions of LSTM-ATT used for generating the experimental results written in this paper, were already optimized through hyperparameter tuning. To resolve this confusion, we added details on the parameters used for our model and the notebooks used for creating the experimental results written on the paper in the appendix section.

---

### Decision · Program_Chairs · 2023-04-09

Invite to archive